# METACASPASE8 (MC8) Is a Crucial Protein in the LSD1-Dependent Cell Death Pathway in Response to Ultraviolet Stress

**DOI:** 10.3390/ijms25063195

**Published:** 2024-03-11

**Authors:** Maciej Jerzy Bernacki, Anna Rusaczonek, Kinga Gołębiewska, Agata Barbara Majewska-Fala, Weronika Czarnocka, Stanisław Mariusz Karpiński

**Affiliations:** 1Institute of Technology and Life Sciences—National Research Institute, Falenty, Al. Hrabska 3, 05-090 Raszyn, Poland; maciej_bernacki@sggw.edu.pl; 2Department of Plant Genetics, Breeding and Biotechnology, Institute of Biology, Warsaw University of Life Sciences, Nowoursynowska Street 159, 02-776 Warsaw, Poland; kinga_golebiewska@sggw.edu.pl (K.G.); agata.barbara.majewska@gmail.com (A.B.M.-F.); 3Department of Botany, Institute of Biology, Warsaw University of Life Sciences, Nowoursynowska 159, 02-776 Warsaw, Poland; anna_rusaczonek@sggw.edu.pl (A.R.); weronika_czarnocka@sggw.edu.pl (W.C.)

**Keywords:** abiotic stress, *Arabidopsis thaliana*, cell death, LSD1, METACASPASES, salicylic acid, reactive oxygen species

## Abstract

LESION-SIMULATING DISEASE1 (LSD1) is one of the well-known cell death regulatory proteins in *Arabidopsis thaliana*. The *lsd1* mutant exhibits runaway cell death (RCD) in response to various biotic and abiotic stresses. The phenotype of the *lsd1* mutant strongly depends on two other proteins, ENHANCED DISEASE SUSCEPTIBILITY 1 (EDS1) and PHYTOALEXIN-DEFICIENT 4 (PAD4) as well as on the synthesis/metabolism/signaling of salicylic acid (SA) and reactive oxygen species (ROS). However, the most interesting aspect of the *lsd1* mutant is its conditional-dependent RCD phenotype, and thus, the defined role and function of LSD1 in the suppression of EDS1 and PAD4 in controlled laboratory conditions is different in comparison to a multivariable field environment. Analysis of the *lsd1* mutant transcriptome in ambient laboratory and field conditions indicated that there were some candidate genes and proteins that might be involved in the regulation of the *lsd1* conditional-dependent RCD phenotype. One of them is METACASPASE 8 (AT1G16420). This type II metacaspase was described as a cell death-positive regulator induced by UV-C irradiation and ROS accumulation. In the double *mc8*/*lsd1* mutant, we discovered reversion of the *lsd1* RCD phenotype in response to UV radiation applied in controlled laboratory conditions. This cell death deregulation observed in the *lsd1* mutant was reverted like in double mutants of *lsd1*/*eds1* and *lsd1*/*pad4.* To summarize, in this work, we demonstrated that MC8 is positively involved in EDS1 and PAD4 conditional-dependent regulation of cell death when LSD1 function is suppressed in *Arabidopsis thaliana*. Thus, we identified a new protein compound of the conditional LSD1-EDS1-PAD4 regulatory hub. We proposed a working model of MC8 involvement in the regulation of cell death and we postulated that MC8 is a crucial protein in this regulatory pathway.

## 1. Introduction

Because of their sessile nature, plants cannot avoid environmental stresses by changing their place of inhabitance. Therefore, in natural environments, plants are constantly exposed to biotic and abiotic stress simultaneously, such as various pathogens, excess/deficiency of light, UV irradiation, drought, cold, heat, or salinity. Throughout the course of evolution, plants have developed many molecular and physiological mechanisms that enable them to simultaneously optimize acclimation and defense responses to variable and adverse environmental conditions [1,2,3]. One of the mechanisms crucial in plants response to stress is cell death (CD). CD is a molecular and physiological process that leads to the selective death of some cells, i.e., mesophyll cells, thus triggering a beneficial immune defense and acclimatory response in other cells [4,5]. In this regard, CD is a specified and highly organized process of the cells’ self-elimination. It plays a crucial role in plant development [6,7], immune defense [8] and acclimatory responses [1,2]. From this point of view, CD is not only the ultimate end of the cell life cycle, but most importantly, it maintains cellular homeostasis in organs and in the whole plant during unfavorable environmental conditions.

Knowledge of the molecular, physiological and genetic mechanisms of plant CD at different levels of complexity (cellular and organismal) was facilitated by the identification of different *Arabidopsis thaliana* mutants exhibiting CD deregulation [9,10,11]. Some of the best-known conditional CD regulators in plants are LESION-SIMULATING DISEASE 1 protein (LSD1), ENHANCED DISEASE SUSCEPTIBILITY 1 (EDS1) and PHYTOALEXIN-DEFICIENT 4 (PAD4). The dysfunctional CD phenotype of the *lsd1* mutant has been broadly studied. This mutant exhibits a runaway cell death (RCD) phenotype which is manifested by the inability to restrict CD propagation if it has been initiated by an external stimulus [3,8,12]. It has been shown that RCD can be induced in the *lsd1* mutant by the following stress factors: high light and photorespiration [13,14], root hypoxia [1,15], drought [16,17], cold [18], UV radiation [16,19] or biotic stresses [20,21]. However, the *lsd1* RCD phenotype is dependent on growing conditions and was not observed when plants were grown in multivariable field conditions [16,22]. Based on many studies, LSD1 is considered a conditional suppressor of CD which is positively regulated by EDS1 and PAD4 and integrates various signaling pathways in response to both biotic and abiotic stresses [3].

Initially, the RCD phenotype of the *lsd1* mutant was linked to the accumulation of superoxide ions produced by plasma membrane-bound NADPH oxidase [8] and only after the other reaction oxygen species (ROS) forms were identified to be involved in RCD phenotype elicitation [13,14,18,23,24]. Since in the *lsd1* mutant the initial level of antioxidative enzyme activity is lower than in the wild type [23,25], LSD1 is regarded as a positive regulator of the enzymatic antioxidant machinery. Another CD-related molecule, which is excessively accumulated in *lsd1*, is salicylic acid (SA) [16,22,23]. It has been found that the artificial blocking of SA accumulation in the *lsd1* mutant prevents RCD induction; therefore, it was proposed that SA accumulation, controlled by LSD1, is essential in triggering CD in response to stress [19].

Dysfunctional overaccumulation of SA in the *lsd1* mutant is caused by the fact that LSD1 can physically interact with proteins involved in SA signaling. It was shown that LSD1 interacts with ENHANCED DISEASE SUSCEPTIBILITY 1 (EDS1), while EDS1 forms complexes with PHYTOALEXIN-DEFICIENT 4 (PAD4) [26,27]. Both EDS1 and PAD4 possess triacylglycerol lipase domains that were originally described as components of gene-mediated and basal disease resistance [28,29,30,31]. EDS1 and PAD4 are crucial for RCD propagation in the *lsd1* mutant, because in the eds1/lsd1 and pad4/lsd1 double mutants, the RCD was inhibited regardless of the stimulus type [21,22,23,32]. Therefore, LSD1 is considered a negative regulator of EDS1- and PAD4-dependent pathways that lead to RCD [3].

From a molecular perspective, an LSD1 protein contains three zinc (Zn)-finger domains that are responsible for DNA/protein binding [33]. LSD1 was proven to be a transcriptional regulator and a scaffold protein [27]. The Zn-finger motifs in LSD1 belong to the C2C2 class that is also present in GaTa1-type transcription factors containing the conserved consensus sequence CxxCRxxLMYxxGaSxVxCxxC [33,34]. Using yeast two-hybrid (Y2H) assay, additional ten putative LSD1-interacting proteins were found, from which one of them was METACASPASE1 (MC1), a positive CD regulator [35].

The METACASAPASES family is interesting in the context of cell death studies. *Arabidopsis thaliana* contains nine METACASAPASES proteins, 1 to 9, that are divided into two groups [36]. Type I metacaspases (MC1, MC2 and MC3) contain zinc-finger domains, while type II (MC4-MC9) do not [36,37]. This family contains both positive and negative CD regulators. While type I metacaspases’ role is relatively well understood [35], the function of the type II subfamily is still largely unknown. In the context of the LSD1-dependent cell death regulation pathway, METACASPASE 8 (MC8) seems to be especially interesting. Its expression level is strongly up-regulated in response to UV-C [38], pathogens [39] and methyl viologen [38]. It was also shown that the recessive mutants in *MC8* exhibit higher resistance to UV-C and ROS treatment [38]. Therefore, we postulated a hypothesis that MC8 is involved in the propagation of RCD in *lsd1* plants and we decided to explore the mutual interdependence of these two proteins in CD regulation.

## 2. Results

### 2.1. MC8 Is Important in the LSD1-Dependent Cell Death Pathway

Because proteins belonging to the METACASPASE family are known to be involved in cell death regulation [35,36,40], we decided to search for MC genes within microarray data published in one of our previous articles [16]. Having compared the fold changes in *MC* genes’ expression level in the *lsd1* mutant and wild-type plants, we observed that only the level of *MC4* and *MC8* were significantly up-regulated (Figure 1A). Interestingly, the expression level of *MC8* was antagonistically regulated in different growing conditions, an ambient laboratory or natural field (Figure 1A). In addition, in *eds1*/*lsd1* and *pad4*/*lsd1* double mutants, which do not exhibit the specific RCD phenotype, the *MC8* fold change was not detected (Appendix A). Data from microarray experiments were confirmed using a real-time PCR (Appendix A). Interestingly, in double *lsd1*/*eds1* and *lsd1*/*pad4* mutants that reverted the *lsd1* RCD phenotype to the wild type, there were not any significant changes in the *MC8* transcript level (Appendix A). This finding together with the reverted cell death in the double *mc8*/*lsd1* mutant in response to UV irradiation (Figure 1B) and the plant phenotype (Figure 1C and Appendix A) allowed us to conclude that MC8 was an important component in the LSD1-dependent conditional cell death regulatory hub. Moreover, we proved that in *Arabidopsis thaliana* wild-type plants, *MC8* expression was strongly up-regulated in response to a combined UV-A + UV-B irradiation episode, while in response to heat or high light stress, there were no differences compared to plants growing in controlled conditions (Appendix A).

### 2.2. MC8 Affects LSD1-Dependent Foliar ROS and SA Levels

In control conditions, we observed a significantly lower foliar H_2_O_2_ level in *lsd1* and *mc8* mutants compared to wild-type plants. The concentration of foliar H_2_O_2_ in the double *mc8*/*lsd1* mutant did not differ significantly from the wild type. After the UV irradiation episode, we observed the highest foliar H_2_O_2_ content in the *lsd1* mutant, while in *mc8*, it was lower than in the wild-type plants. The double *mc8*/*lsd1* mutant did not differ from the wild type in terms of H_2_O_2_ content, while the *mc8* mutant demonstrated the lowest H_2_O_2_ content after UV stress (Figure 2A). Salicylic acid is an important cell death signaling molecule and the RCD phenotype of *lsd1* is related to high foliar SA content [19,20]. Because of this fact, we performed an analysis of SA content in foliar tissues of tested *Arabidopsis thaliana* mutants. In control conditions, the *lsd1* mutant did not differ much from the wild type, while the *mc8* plants exhibited an insignificantly lower SA content. After UV stress, foliar SA levels were strongly increased in *lsd1*, but in *mc8* and in double *lsd1*/*mc8* mutants, they were increased similarly like in the wild-type plants (Figure 2B).

### 2.3. High PR5 Gene Expression Level in lsd1 Is Reverted by the Mutation in MC8

*PR* genes’ expression is strongly up-regulated in the *lsd1* mutant [19] and their expression is related to SA levels in plant tissues [41,42]. Therefore, we decided to check the expression of *PR1*, *PR2* and *PR5* in all lines being investigated in this study. In control conditions, we found similarly low expression levels of *PR1* and *PR2* in all tested genotypes. The *PR5* expression level was differentiated, and its expression was significantly higher in the *lsd1* mutant in control conditions in comparison to the wild type and other tested mutants (Figure 3A–C). After UV irradiation, *PR1* expression was higher in all tested genotypes when compared to control conditions. However, the *PR1* transcript was drastically increased in *lsd1* and *mc8*/*lsd1* mutants (Figure 3A). Upon UV stress, the *PR2* expression level was significantly higher in *lsd1* and *mc8*/*lsd1* mutants (Figure 3B). However, the expression level of *PR5* after UV stress was significantly higher only in the *lsd1* mutant when compared to the wild type (Figure 3C).

### 2.4. PR Proteins Are Degraded in Response to UV Stress

Despite the fact that we found a very strong up-regulation of *PR1* gene expression in *lsd1* and *mc8*/*lsd1* mutants in stress conditions (Figure 3A), we did not observe a higher PR1 protein level. In fact, the PR1 protein level was similar in wild type plants and decreased in other tested genotypes after UV stress, in comparison to control conditions (Figure 4A). Even though the expression of the *PR2* gene was strongly up-regulated in *lsd1* and *mc8*/*lsd1* mutants in response to UV irradiation, in *lsd1* and in mc8/*lsd1*, the level of the PR2 protein decreased (Figure 4B). The PR5 protein was present in all of the tested genotypes; however, the higher amount of this protein was found in *mc8* and *lsd1*/*mc8* mutants in control conditions. The level of PR5 protein decreased in response to stress in all of the tested genotypes (Figure 4C). The experiment was performed in two independent biological replicates (Appendix A).

### 2.5. Prediction of MC8 Interaction

The protein–protein interaction of LSD1 is well known, and there is no information about any LSD1-MC8 interaction [27]. There is no prediction of the potential interaction of MC8 with LSD1, EDS1 and PAD4 nor of any other proteins which were previously described as LSD1, EDS1 and PAD4 interactors. This indicates the lack of protein–protein interaction with the above-mentioned CD-regulating proteins. Moreover, it shows that the analyzed pathway is dependent on hormones and ROS, which are regulated by LSD1/EDS1/PAD4 proteins (Appendix A and Appendix A).

## 3. Discussion

Plants are constantly exposed to various types of environmental stresses and because of that, they have developed many mechanisms and pathways to mitigate the effects of unfavorable conditions. One of the mechanisms activated during plant response to stress is programed cell death (PCD), a very old evolutionary mechanism present in all multicellular living organisms [43,44,45]. The role of PCD in plant response to stresses is basically the elimination of affected and/or older cells in order to induce higher tolerance and better acclimation in the other cells and maintenance of cellular homeostasis within an organism [46,47]. It is well known that this process is under conditional control of the LSD1 protein as well as EDS1 and PAD4, which are negative and positive cell death regulators, respectively [16,22,23,26,48]. Reversal of the *lsd1* RCD phenotype is nothing new. This effect was obtained by mutation in *EDS1* or *PAD4* genes in the *lsd1* mutant background, because these proteins physically interact with each other, forming a trimmer [26,27]. When there is no LSD1 protein present (*lsd1* mutant) or it is at a lower level (in field conditions), the EDS1 and PAD4 proteins are active, and when there is only EDS1 (*lsd1*/*pad4* mutant) present, it cannot induce a PR pathway alone [21,22,23,49]. Another way to revert the *lsd1* phenotype is to disable the chloroplast signal recognition particle cpSRP 43 protein (encoded by the *CAO* gene) [32]. CAO is involved in the regulation of light-harvesting antenna size and non-photochemical quenching of absorbed energy in excess by photosystem II [50]. In the *lsd1*/*cao* double mutant, there is a reversion of the RCD phenotype due to its reduced ability to absorb light energy [3,32]. Thus, LSD1 is a negative regulator of PCD and photorespiration and a positive regulator of the antioxidant system [3]. Another piece of evidence to support chloroplast retrograde signaling mediated by cytoplasmic LSD1/EDS1/PAD4 proteins is presented in the *lsd1/ex1* mutant which did not exhibit the RCD phenotype [51]. EXECUTER1 (EX1) is involved in singlet oxygen chloroplast retrograde signaling. ROS do not act alone in plant cells and the relation between SA and ROS in plant stress response is well known [22]. Because of this, deregulation in SA synthesis/metabolism also leads to reversion of the *lsd1* phenotype in mutants such as *lsd1*/*sid2* or in transgenic plants like *lsd1*/NahG [19,20]. Ours results and those of previous studies strongly suggest that a unified genetic and molecular system for the regulation of biotic and abiotic stress responses and cross-tolerance has evolved in plants [52]. Ethylene is also involved in conditional regulation by LSD1/EDS1/PAD4 PCD signaling, since impairment of this signaling pathway in the *ein2*/*lsd1* double mutant also leads to reversion of the RCD *lsd1* phenotype [2]. Interestingly, the *lsd1* mutant, when grown in field conditions, did not differ from the wild type in size, visually or in seed yield [16,22]. In ambient laboratory conditions, *lsd1* mutant plants are much smaller and produce much fewer seeds (fourfold lower seed yield) than the wild-type or double *eds1*/*lsd1* and *pad4*/*lsd1* plants. While all currently known ways to revert the *lsd1* RCD phenotype are related to changes in ROS, SA or ethylene signaling pathways or to LSD1-interactor proteins, in this study, we found a new protein, MC8, which, according to the current knowledge [38,53] and bioinformatic prediction (Appendix A and Appendix A), does not interact with LSD1 and is not involved in SA, ROS and ET synthesis/metabolism, which suggests its functions as a signal receiver (ROS/hormonal) and as the final enforcer of LSD1-dependent cell death. During the analysis of our previously performed microarray [16], in the context of genes from the METACASPASE family, we found a significant positive fold change in *MC8* in laboratory growing conditions (inducing RCD) and a significant negative fold change in field conditions (not inducing RCD). Moreover, in double *lsd1*/*eds1* and *lsd1*/*pad4* mutants with well-known reversion of the *lsd1* RCD phenotype [22,23], there was a non-positive fold change in *MC8*, which indicates that MC8 acts upstream of the LSD1/EDS1/PAD4 trimer. These results were confirmed by studies on the *mc8*/*lsd1* mutant. A double mutant exhibited a completely reverted RCD phenotype. This is probably caused by the fact that MC8 transcription is ROS-dependent, which was experimentally demonstrated by exogenous H_2_O_2_ treatment and by the induction ROS synthesis by of UV-C stress [22,38]. In conclusion, we can assume that MC8 is an important protein in the conditional LSD1/EDS1/PAD4-dependent cell death regulatory pathway. MC8 most probably is not related to SA and ROS synthesis or metabolism since in the *lsd1*/*mc8* double mutant, we found lower or similar foliar levels of SA and H_2_O_2_ to those observed in wild-type plants. It is probably involved in the very early stage of metacaspases cascades of PCD and/or the inhibition of the second burst of the ROS/SA [44,54] wave from cell organelles undergoing PCD, which was observed in the RCD of the *lsd1* mutant [23] or generally while PCD progressed in wild-type plants [44,55]. Inhibition of gene expression from the *PR* family was previously proposed as one of the reasons for the inhibition of RCD in *lsd1/eds1*, *pad4/lsd1*, *sid2/lsd1*, etc. [19,49]. However, in the *lsd1/mc8* mutant, EDS1 and PAD4 were not inhibited by LSD1. This led to the induction of *PR1* and *PR2* expression in *lsd1/mc8* on a similar level as in the *lsd1* mutant. The exception is *PR5* and its higher expression, which was observed only in the *lsd1* mutant but not in *lsd1/mc8*. It could be caused by the fact that SA and ROS levels were slightly, but statistically significantly higher in *lsd1* compared to *lsd1/mc8*. There are some alternatives to the SA-related signaling pathway [56] and there is a reverse correlation between H_2_O_2_ steady-state concentration and *PR5* gene expression [57]. However, the number of PR proteins appear to be different to the *PR* genes’ transcription level. In general, in all of the mutants and in the wild-type plants, 24 h after UV-A + B stress, we found lower levels of PR1, PR2 and PR5. This is opposite to biotic stress, where the number of PR proteins were higher after pathogen inoculation [58]. This indicates that in response to abiotic stress (UV stress), a *PR* gene family is regulated differently on transcription and translation levels. However, this requires further research. Based on the current knowledge and on our results, we propose a model of LSD1 and MC8 interdependence in plant PCD regulation in response to UV-A + B irradiation (Figure 5). The UV episode is comprehended mostly by chloroplasts and provokes changes in the quantum-redox status of the photosynthetic electron transport chain components [3,59,60,61,62]. It leads to ROS overproduction [63], which consequently leads to increased synthesis of SA [3]. Both ROS and phytohormones act as a signal for LSD1 inhibition, which leads to EDS1 and PAD4 increasing in activity and then to the induction of cell death. Nevertheless, in other cells, LSD1 is induced, thus inhibiting EDS1 and PAD4 and preventing PCD spreading. Meanwhile, in the *lsd1* mutant, there is no LSD1, thus EDS1 and PAD4 are hyperactive and act as inhibitors of the antioxidant system [3] and are important in SA synthesis [64,65]. This altogether leads to RCD induction [66]. However, in *lsd1*/*mc8*, all ROS/SA/ET signaling is still deregulated because of the lack of functional LSD1 and the lack of LSD1/EDS1/PAD4 trimers, thus EDS1 and PAD4 should induce the RCD phenotype, but such phenotype in *lsd1/mc8* is not observed after UV stress. This is probably because MC8 is a receiver of the above-mentioned signals and MC8 acts upstream of all of the above-described proteins and thus is the ultimate executor of the run-away cell death process.

## 4. Materials and Methods

### 4.1. Plant Material

In this study, *Arabidopsis thaliana* mutants, *lsd1*, *mc8* and *mc8/lsd1*, and wild-type plants (Ws-0) were used. All used mutants were of Wassilewskija (Ws-0) background. Wild-type and *lsd1* seeds were already available in our lab, while *mc8* seeds were provided by Dr. Patrick Gallois (Faculty of Biology, Medicine and Health, University of Manchester, Manchester, UK). In order to obtain a double mutant, we crossed *lsd1* and *mc8*. A double mutant was obtained via selection in T3 generation. Correctness of the crossing was checked by PCR (Appendix A) and by RT-PCR (Appendix A). All primers used in the study are attached in the Appendix A.

### 4.2. Growing Conditions

For all experiments described in this study, plants were grown in a walk-in-type growing chamber (Siemens, München, Germany) under the following conditions: 8/16 h photoperiod, photosynthetic photon flux density of 80 μmol photons m^−2^·s^−1^, air humidity of 50% and day/night temperature of 20/18 °C.

### 4.3. Stress Induction

For the determination of different stress impacts on *MC8* expression levels, wild-type plants were treated with heat, light and UV-A + B using the following methods. For UV A + B stress application, the UV 500 Crosslinker (Hoefer Pharmacia Biotech, San Francisco, CA, USA) was used. It was equipped with three UV-B lamps (type G8T5E, Sankyo Denki, peak wavelength 306 nm) and two UV-A lamps (type TL8WBLB, Philips, Tokyo, Japan, peak wavelength 365 nm). *Arabidopsis thaliana* plants were exposed to a single irradiation dose of 1500 mJ·cm^−2^. Light treatment was performed using a white LED panel with the emission of white light (with 1500 μmol photons m^−2^·s^−1^) (Photon System Instrument, Brno, Czech Republic) for 2 h. For heat stress treatment, plants were incubated in 40 °C for two hours in the laboratory incubator MOV-212s (Philips, Tokyo, Japan). All analyses described in this study were performed 24 h after stress application. After the stress episode, plants were put back in the growing chamber in the same conditions as they were grown in before. For future experiments, UV A + B were chosen, and wild-type plants and all mutants used in this study were exposed to a single irradiation dose of 1500 mJ·cm^−2^.

### 4.4. Ion Leakage Measurement

Ion leakage was determined as described before [19,44].

### 4.5. Determination of H_2_O_2_ and SA Contents

The concentration of hydrogen peroxide (H_2_O_2_) was assessed according to the method described before [67], with slight adjustments. A total of 50–100 mg of frozen tissue was homogenized in a TissueLyser LT (Qiagen, Venlo, The Netherlands) for 5 min at 50 Hz and 4 °C, using 300 µL of cold 0.1% trichloroacetic acid (TCA), then centrifuged at 13,000 rpm for 15 min. The resulting supernatant was combined with 10 mM potassium phosphate buffer (pH 7.0) and 1 M potassium iodide (KI) at a 1:1:2 (*v*:*v*:*v*) ratio. The absorbance was measured at 390 nm using a microplate reader, Multiscan GO (Thermo Scientific, Waltham, MA, USA), and the H_2_O_2_ concentration was determined using an appropriate standard curve. Results were quantified and expressed as micromoles of H_2_O_2_ per 100 mg of fresh weight.

The determination of salicylic acid (SA) followed the protocol described before [68], with the utilization of 2-methoxybenzoic acid (oANI) and 3-hydroxybenzoic acid (pHBA) as internal standards. Salicylic acid was separated using a Luna 5uC18(2)100A150x4.6mm column (Phenomenex, Torrance, CA, USA) at 30 °C for 15 min, employing a Shimadzu HPLC System (Shimadzu, Kyoto, Japan). A low-pressure gradient system was used, utilizing a 20 mM phosphate buffer (pH 2.5; adjusted with 8 M HCl) and acetonitrile (75:25; *v*/*v*) at a flow rate of 1 mL per minute. Results were quantified and expressed as micrograms of SA per gram of fresh weight.

### 4.6. RNA Isolation, cDNA Synthesis and RT-PCR Analysis

RNA was extracted from frozen tissue previously stored in −80 °C. For RNA isolation, the GeneMaTRIX Universal RNA Purification Kit (EURX, Gdańsk, Poland) with an additional step of on-column DNase digestion was used. RNA concentration and quality were assessed using a spectrometer (Eppendorf, Hamburg, Germany). RNA quality was controlled by electrophoretic separation in 1% agarose gel. cDNa synthesis was performed for equimolar RNA amounts of each sample using a High Capacity cDNa Reverse Transcription Kit (Thermo Fisher Scientific). qPCRs were performed in three technical repetitions for each of the three biological replicates using the Power SYBR Green PCR Master Mix and the aBI 7500 Fast Real-Time PCR System (Thermo Fisher Scientific, Waltham, MA, USA). Two reference genes were used: 5-FORMYLTETRAHYDROFOLATE CYCLOLIGASE (5-FCL, AT5G13050) and PROTEIN PHOSPHATASE 2A SUBUNIT A2 (PP2AA2, aT3G25800).

### 4.7. Protein Extraction and Western Blot Analysis

Total plant protein extraction was performed as previously described [69], with modifications. Proteins were extracted from 100 mg of ground leaf tissue. The powder was resuspended in 500 μL of 2× Leammli buffer (4% SDS, 20% glycerol, 0.12 M Tris-HCl pH 7.0, 0.02% bromophenol blue and 0.7 M β-mercaptoethanol) and incubated for 10 min at 95 °C, followed by incubation for 10 min in ice [70]. The resuspended samples were centrifuged at 12,000× *g* for 5 min at 4 °C and the supernatants were used for further steps. Total protein concentration was determined using the RC-DC protein assay kit II (Bio-Rad, Hercules, CA, USA, 5000122). A total of 50 μg of total protein extract was used for 12% SDS-PAGE. Next, the proteins were electrotransferred to an Immobilon P PVDF membrane (Merck, Darmstadt, Germany) using the Trans-Blot Turbo Transfer System (Bio-Rad). The membrane was blocked in 2% skim milk in TBS-T buffer (1× Tris-buffered Saline, 0.1% Tween-20) for at least 1 h and incubated with primary antibodies (diluted at a ratio of 1:10,000, 2% skim milk in TBS-T buffer)—PR1 (AS10 687, Agrisera, Vännäs, Sweden) for 1 h at room temperature, and PR2 (AS12 2366, Agrisera) and PR5 (AS12 2373, Arisera) overnight at 4 °C. Incubation with the goat anti-rabbit horseradish peroxidase-conjugated secondary antibodies (Thermofisher, QG221919; diluted at a ratio of 1:10,000) was performed for 1 h at room temperature. Protein bands were immunodetected using SuperSignal West Dura Extended Duration Substrate (Thermo Scientific, 34075) according to the manufacturer’s recommendations, visualized with the ChemiDoc XRS+ System (Bio-Rad) and analyzed with ImageLab Software 5.2.1 (Bio-Rad). Total protein staining of membranes was conducted as described previously [71].

## Figures and Tables

**Figure 1 ijms-25-03195-f001:**
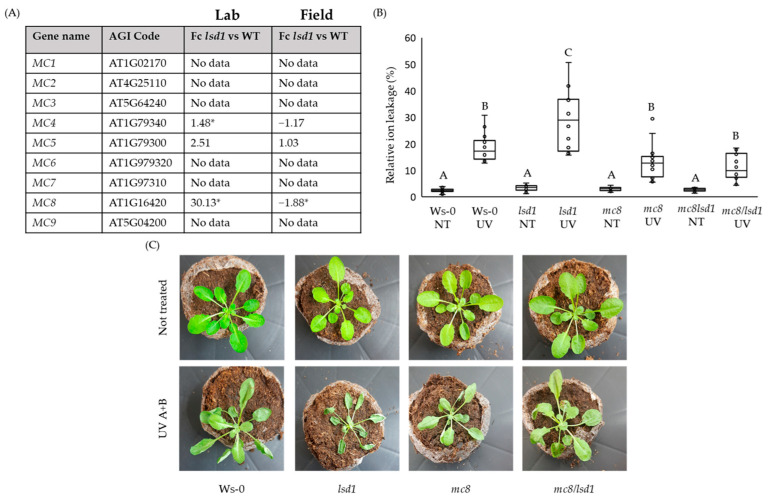
The role of METACASPASE 8 in conditional LSD1-dependent cell death regulation. (**A**) Analysis of transcriptomic data [16] in the context of expression changes in genes encoding METACACASPES family proteins, and in *lsd1* mutant grown in ambient laboratory conditions (Lab) or in natural field conditions (Field), significant changes are marked with asterisks. FC—fold change. (**B**) Level of measured foliar ion leakage (manifesting the cell death level) for plants grown in control conditions and exposed to episode of UV-A + UV-B irradiation. Within a subgraph, values sharing the same letters are not significantly different from each other (*p* > 0.001) (n = 10–15). (**C**) Pictures of *Arabidopsis thaliana* rosettes from control conditions (upper row) and after UV-A + UV-B irradiation incidents (bottom row).

**Figure 2 ijms-25-03195-f002:**
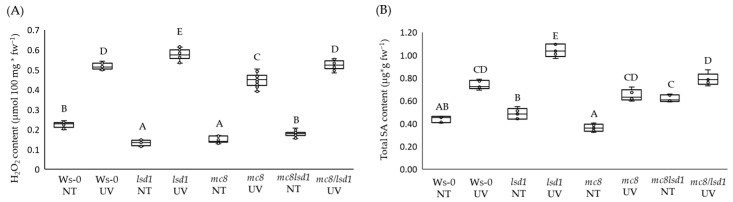
Foliar (**A**) H_2_O_2_ and (**B**) salicylic acid levels in control conditions and 24 h after UV irradiation. Within a subgraph, values sharing common letters are not significantly different from each other (*p* > 0.001) (n = 6).

**Figure 3 ijms-25-03195-f003:**
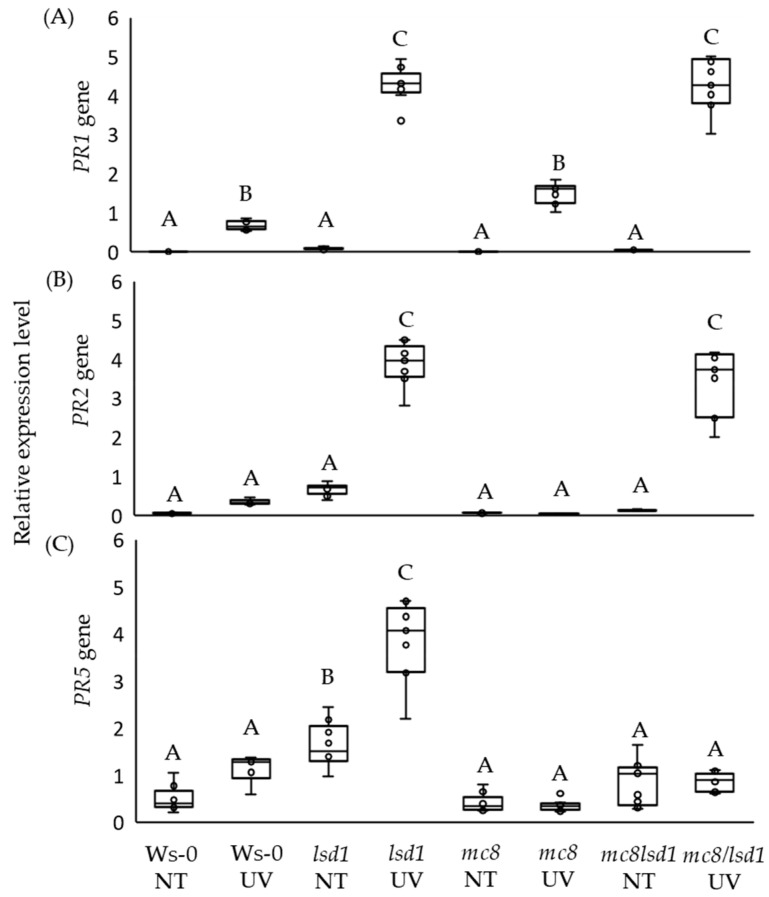
*PR* genes’ relative expression levels in non-stress ambient laboratory conditions and in response to UV irradiation episode. (**A**) *PR1*, (**B**) *PR2* and (**C**) *PR5* expression levels in tested mutants in control conditions and 24 h after UV irradiation. Within a subgraph, values sharing common letters are not significantly different from each other (*p* > 0.001) (n = 9).

**Figure 4 ijms-25-03195-f004:**
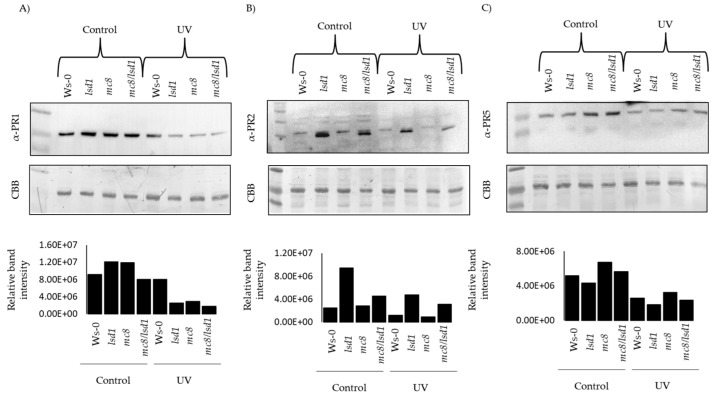
Level of PR proteins in response to UV irradiation. Western blot analysis of (**A**) PR1, (**B**) PR2 and (**C**) PR5 proteins in all tested genotypes in control conditions and 24 h after UV irradiation. The graphs show relative intensity of the bands (presented on gels) for each tested protein. As a loading control, CBB staining was used.

**Figure 5 ijms-25-03195-f005:**
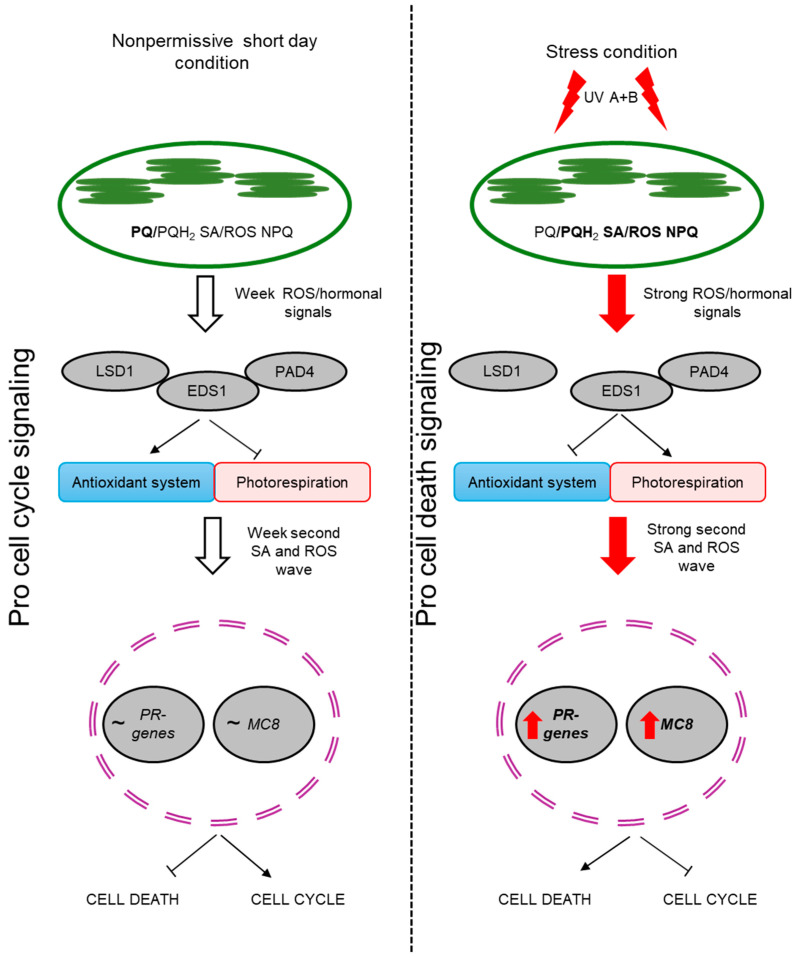
Proposed/hypothetical model of LSD1 and MC8 interdependence in plant cell death regulation in response to UV-A + B irradiation. We propose that MC8 is more important in the cell death pathway dependent on LSD1 than the proposed EDS1 and PAD4 proteins (there are more details in the last paragraph of the discussion). The arrow is the inducing action; The blunt-headed arrow is braking; The red arrow next to the gene names is increased expression; The tilde sign is an expression without change.

## Data Availability

All of the data generated or analyzed during this study are included in the published article and its Appendix A.

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
