# Peer review of "METACASPASE8 (MC8) Is a Crucial Protein in the LSD1-Dependent Cell Death Pathway in Response to Ultraviolet Stress"

_ijms, 2024, doi:10.3390/ijms25063195_

Round 1
Reviewer 1 Report
Comments and Suggestions for Authors
Comments on the Quality of English LanguageMany sentences have not been carefully proofread which leads to syntax errors that sometimes make the manuscript difficult to read, especially in the Discussion section.
Author Response
Dear Reviewer 1
We kindly thank you for all your comments and remarks. We have studied them with particular attention and carefully introduced them in the text. All changes that will be introduced in the text will be marked in a legible way so that you can easily read them. Replies to your comments are marked in red. Marked in yellow and green refer to the comments of other reviewers In addition, the entire text was checked and corrected for linguistic correctness by an external editor.
1 – References 4 and 5 refer to programmed cell death in animals and are not adequate.
According to your comment questionable literary items have been removed from the text
2 – Line 30: instead of “demonstrating a hypothetical model”, “proposing a model” is more accurate.
The model we present is the result of our research and references to current scientific knowledge. We also post it based on data from databases such as STRING, in which part of the data is based on bioinformatic predictions. We have changed your proposed sentence to: We proposed a working model of MC8 involvement in regulation of cell death and we postulate that MC8 is a crucial protein on this regulatory pathway. Is marked yellow in manuscript.
3 – Line 114: “on at”, one of the many syntax error in the manuscript ; line 116: unclear sentence, please rephrase ; line 122: “prove” or “proved” instead of “proof”
We did some correction. Are marked in manuscript (yellow color)
4 – Fig. 1A is barely readable, please increase size; Lab/Field is not mentioned in the Table, please add.
Figure 1 has been corrected. The font in the table has been increased by two points, the description LAB, FIELD has been added. The corrected figure is in the new version of the manuscript
5 – Fig. 3: the Y axis legend is incorrect (PR11-PR15?)
This obvious error has been corrected. The corrected figure is in the new version of the manuscript
6 – Line 178: check sentence; line 182: “replications” or “replicates”
The sentences have been corrected
7 – Fig. 4: first sentence (title) in the legend is incorrect.
The title of figure 4 has been corrected. Marked yellow.
8 – Line 189-190: unintelligible sentence… ; I agree for presenting the 2.5 section data in
Supplementary data but the text of the section should be more descriptive; the authors
should specify why they conclude that MC8 interacts with AT3G24530 (misspelled in the
text) and AT3G03370. The second is only connected with the first and if authors consider
that AT3G03370 is connected with MC8 through its interaction with AT3G24530, MC2 is in
the same situation.
Figure 5 has been corrected, a record of genotypes has been added under the band intensity charts. The sentence has been corrected to make sense
Section 4.5: the method is not detailed in the reference provided (please number it) which itself refers to a previous reference… The authors should provide the method in the text.
The methodology for SA and ROS has been described in detail

Reviewer 2 Report
Comments and Suggestions for Authors
The authors have done an excellent job constructing a narrative that connects the genetic interaction between LSD1 and AtMC8 with cell death induced by UV stress. However, it's essential to note limitations in attributing the link solely to MC8. Given the existence of numerous AtMCs, incorporating the well-known functions of AtMC1 and AtMC2 could provide valuable insights for this study. If there are no results regarding genetic and protein interactions, the paper may fall short in fully demonstrating its potential contributions.
The authors are encouraged to present additional data, specifically incorporating experiments like Y2H or CoIP, to ascertain the impact of LSD1-AtMC8 protein interaction and the interaction of LSD1 with other AtMCs on the LSD1-AtMC8 complex. Additionally, for data visualization, it is recommended to utilize tools such as the R-program or graphic software to generate more visually stable figures. While this journal does not impose extra costs for color figures, it is advisable to ensure that figures are designed in a format that enhances clarity and visibility.
The PR genes utilized by the authors are marker genes with rapidly increasing gene levels. Considering their quick response, the significance of examining them within a 24-hour timeframe may be questioned. It is suggested to assess the expression of PR genes post-UV treatment, as a clear phenotype in their expression could emerge within 30 minutes to 1 hour. Moreover, reliance on PR genes, which are markers for biotic stress, for UV-induced cell death might present an issue. Instead, it is recommended to include cell death markers specific to environmental stress in the study."
In Figure 4, what significance do authors believe the experiment confirming the level of PR protein at 24 hours demonstrates? Is there a potential correlation between the decrease in the level of PR protein due to UV stress and the observed increase in cell death in mutant plants? It might be advisable to verify this using a different antibody.
How does AtMC8 regulate the expression of PR proteins? The authors are encouraged to provide insights into whether AtMC8 is directly involved in processing PR proteins or if it regulates them by modulating the activity of other proteins.
The authors should incorporate direct experimental data on the association of LSD1-EDS1-PAD4 in UV-induced apoptosis. While there are well-established studies linking this complex to drought stress or disease resistance, there appears to be a gap in research concerning UV stress.
The minor point is to distinguish between genes and proteins. ‘PR gene and PR protein’
Comments on the Quality of English LanguageThe minor point is to distinguish between genes and proteins. ‘PR gene and PR protein’
Author Response
Dear Reviewer 2
We kindly thank you for all your comments and remarks. We have studied them with particular attention and carefully introduced them in the text. All changes that will be introduced in the text will be marked in a legible way so that you can easily read them. Replies to your comments are marked in red. Marked in yellow and green refer to the comments of other reviewers In addition, the entire text was checked and corrected for linguistic correctness by an external editor.
Our reply to yours comments
The authors are encouraged to present additional data, specifically incorporating experiments like Y2H or CoIP, to ascertain the impact of LSD1-AtMC8 protein interaction and the interaction of LSD1 with other AtMCs on the LSD1-AtMC8 complex
We realize that we have not conducted such studies using MC8 as a bait, however there are article from our team about LSD1 interaction (Czarnocka et al., 2017) in with we shown that there is no LSD1-MC8 interaction. We add sentence about it in the manuscript and we add citation od this paper. In addition we would like to explain that MC is a protease involved into cell death execution after induction of point of no return in a massive (cascade) activation of various methacaspases and other proteases involved in PCD so it is rather not interacting with LSD but rather digesting it.
Additionally, for data visualization, it is recommended to utilize tools such as the R-program or graphic software to generate more visually stable figures. While this journal does not impose extra costs for color figures, it is advisable to ensure that figures are designed in a format that enhances clarity and visibility
In response to your comment and the comments of other reviewers, we have improved the readability of the figures. Fonts and the quality of exported TIFF files have been improved.
The PR genes utilized by the authors are marker genes with rapidly increasing gene levels. Considering their quick response, the significance of examining them within a 24-hour timeframe may be questioned. It is suggested to assess the expression of PR genes post-UV treatment, as a clear phenotype in their expression could emerge within 30 minutes to 1 hour. Moreover, reliance on PR genes, which are markers for biotic stress, for UV-induced cell death might present an issue. Instead, it is recommended to include cell death markers specific to environmental stress in the study."
In our previous studies, we showed that genes from the PR gene family are a very good markers for UV stress response after 24 hours (Bernacki et al., 2021) and this result is repeatable in the example of these studies. Moreover, we would like to point out that high expression of PR genes after a UV episode is visible even after 54 hours after stress. Massive PCD induction after UV stress is observed 2-3 days after stress application ((A -H -Mackerness et al., 2001; Surplus et al., 1998). So 24 h is a good point of no return in term of foliar PCD development.
In Figure 4, what significance do authors believe the experiment confirming the level of PR protein at 24 hours demonstrates? Is there a potential correlation between the decrease in the level of PR protein due to UV stress and the observed increase in cell death in mutant plants? It might be advisable to verify this using a different antibody.
We found no correlation between the amount of cell death PR protein in the tested mutants. In the plants observed after UV stress, we generally observed lower content of PR proteins than in control plants. We tried to find literature data on PR proteins level after stress application. In general, it is well known that the content of this protein increases during SAR and HR, but there is not much information about the content of PR protein under abiotic stresses like UV or High light. In our opinion 30 min or 24 h after stress will not change the cell destiny in lsd1 mutant it’ll induce massive PCD – runaway cell death while in double lsd1/mc8 mutant runaway cell death phenotype is reverted to wild type plant cell death phenotype like double lasd1/eds1 and lsd1/pad4.
How does AtMC8 regulate the expression of PR proteins? The authors are encouraged to provide insights into whether AtMC8 is directly involved in processing PR proteins or if it regulates them by modulating the activity of other proteins.
This is misunderstanding LSD1, EDS1 and PAD4 are transcription regulators and might regulate PR gene expression directly by binding to the promoter cis regulatory element. MC is a protease (plant metacaspases) which digesting cell content after induction of point of no return for PCD. We tested the PR genes and proteins more to illustrate the fact that the plants are under stress. We do not believe that MC8 is in any way involved in the regulation of PR gene expression or in posttranslational processing of PR proteins. PR proteins are strongly associated with SA/ROS regulated for example by LSD1, EDS1 or PAD4 signaling, and MC8 is involved in execution of PCD (cell digestion).
A -H -Mackerness, S., John, C.F., Jordan, B., Thomas, B., 2001. Early signaling components in ultraviolet-B responses: distinct roles for different reactive oxygen species and nitric oxide. FEBS Lett. 489, 237–242.
Bernacki, M.J., Rusaczonek, A., Czarnocka, W., Karpiński, S., 2021. Salicylic Acid Accumulation Controlled by LSD1 Is Essential in Triggering Cell Death in Response to Abiotic Stress. Cells 10, 962. https://doi.org/10.3390/cells10040962
Czarnocka, W., Van Der Kelen, K., Willems, P., SzechyÅ„ska-Hebda, M., Shahnejat-Bushehri, S., Balazadeh, S., Rusaczonek, A., Mueller-Roeber, B., Van Breusegem, F., KarpiÅ„ski, S., 2017. The dual role of LESION SIMULATING DISEASE 1 as a condition-dependent scaffold protein and transcription regulator. Plant Cell Environ. 40, 2644–2662. https://doi.org/10.1111/pce.12994
Surplus, S.L., Jordan, B.R., Murphy, A.M., Carr, J.P., Thomas, B., -Mackerness, S. a.-H., 1998. Ultraviolet-B-induced responses in Arabidopsis thaliana: role of salicylic acid and reactive oxygen species in the regulation of transcripts encoding photosynthetic and acidic pathogenesis-related proteins. Plant, Cell & Environment 21, 685–694. https://doi.org/10.1046/j.1365-3040.1998.00325.x

Reviewer 3 Report
Comments and Suggestions for Authors
The manuscript describes why “METACASPASE8 (MC8) is a crucial protein on LSD1-dependent cell death pathway in response to ultraviolet stress”.
The text is generally not well-written, particularly the Discussion Section.
Other revisions include:
- Ls.111-2: Remove the internet address.
- L.114: “…of at MC8…”?
- L.204: “…this two proteins…”?’
- L.205: Trimmer?
- Ls.216-8: “Thus ROS 216 and SA are strongly interdependet each other [24] and also deregulation in SA synthe-217 sis/metabolism lead to reversion of the lsd1 phenotype in mutant such as lsd1/sid2 or trans-218 genic plants lsd1/NahG [21,22].’ Not clear.
- Ls.224-234: “What is interesting lsd1 mutant when growth … suggests that could be the final enforcer of LSD1-dependent cell death.” Not clear.
- Ls.256-60: “The exception is … concentration and the PR5 gene 259 expression [59].” Not clear.
- Ls.271-4: “All this ROS and phytohormones act for LSD1 inhibition … in SA and synthesis [66,67].” Not clear.
- Ls.278-80: “It is probably because …the run-away cell death process.’ Not clear.
- Ls.307-8: “Wild type plants were irradiated with 1500 μmol photons m−2·s−1 for 2 hours.” Please, explain: a) the applied dose, b) why in wild type plants, c) which dose was applied to the rest and for how long, and d) add suitable Refs to justify the doses.
- Section 4.7: References are needed.
Comments on the Quality of English Language
The text is generally not well-written, particularly the Discussion Section.
Author Response
Dear Reviewer 3
We kindly thank you for all your comments and remarks. We have studied them with particular attention and carefully introduced them in the text. All changes that will be introduced in the text will be marked in a legible way so that you can easily read them. Replies to your comments are marked in green. Marked in yellow and red refer to the comments of other reviewers In addition, the entire text was checked and corrected for linguistic correctness by an external editor.
Our reply to yours comments
Ls.111-2: Remove the internet address.
We have removed the www address
L.114: “…of at MC8…”?
The sentence has been corrected (marked in yellow in the text because there was also reviewer's comment 1)
L.204: “…this two proteins…”?’
We have corrected this sentence
L.205: Trimmer?
We have corrected this sentence
Ls.216-8: “Thus ROS 216 and SA are strongly interdependet each other [24] and also deregulation in SA synthe-217 sis/metabolism lead to reversion of the lsd1 phenotype in mutant such as lsd1/sid2 or trans-218 genic plants lsd1/NahG [21,22].’ Not clear.
We have corrected this sentence
Ls.224-234: “What is interesting lsd1 mutant when growth … suggests that could be the final enforcer of LSD1-dependent cell death.” Not clear.
We have corrected this sentence
Ls.256-60: “The exception is … concentration and the PR5 gene 259 expression [59].” Not clear.
We have corrected this sentence
Ls.271-4: “All this ROS and phytohormones act for LSD1 inhibition … in SA and synthesis [66,67].” Not clear.
We have corrected this sentence
Ls.278-80: “It is probably because …the run-away cell death process.’ Not clear.
We have corrected this sentence
Ls.307-8: “Wild type plants were irradiated with 1500 μmol photons m−2·s−1 for 2 hours.” Please, explain: a) the applied dose, b) why in wild type plants, c) which dose was applied to the rest and for how long, and d) add suitable Refs to justify the doses.
This was an error in the text, of course the point was that all genotypes used in the experiments were exposed to UV AB radiation. We have corrected this.
Section 4.7: References are needed.
Reference were added

Round 2
Reviewer 3 Report
Comments and Suggestions for Authors
No comments
Comments on the Quality of English LanguageMinor editing of English language required
Author Response
Dear reviewer
Thank you very much for all your comments. We have made every effort to implement them, and we strongly believe that it has made our manuscript much better.
The changes have been made in track changes mode, so they will be clearly visible to you.
Kind regards
Stanisław Karpiński
